# In Vitro Properties and Virulence of Contemporary Recombinant Influenza B Viruses Harboring Mutations of Cross-Resistance to Neuraminidase Inhibitors

**DOI:** 10.3390/v11010006

**Published:** 2018-12-22

**Authors:** Clément Fage, Yacine Abed, Liva Checkmahomed, Marie-Christine Venable, Guy Boivin

**Affiliations:** Research Center in Infectious Diseases of the CHUQ-CHUL and Laval University, 2705 Boulevard Laurier, Québec City, QC G1V 4G2, Canada; Clement.Fage2@crchudequebec.ulaval.ca (C.F.); yacine.abed@crchudequebec.ulaval.ca (Y.A.); liva.checkmahomed@crchudequebec.ulaval.ca (L.C.); marie-christine.venable@crchudequebec.ulaval.ca (M.-C.V.)

**Keywords:** influenza B, resistance, fitness, mouse model and neuraminidase mutation

## Abstract

Three neuraminidase inhibitors (NAIs: Oseltamivir, zanamivir and peramivir) are currently approved in many countries for the treatment of influenza A and B infections. The emergence of influenza B viruses (IBVs) containing mutations of cross-resistance to these NAIs constitutes a serious clinical threat. Herein, we used a reverse genetics system for the current B/Phuket/3073/2013 vaccine strain to investigate the impact on in vitro properties and virulence of H136N, R152K, D198E/N, I222T and N294S NA substitutions (N2 numbering), reported by the World Health Organization (WHO) as clinical markers of reduced or highly-reduced inhibition (RI/HRI) to multiple NAIs. Recombinant viruses were tested by NA inhibition assays. Their replicative capacity and virulence were evaluated in ST6GalI-MDCK cells and BALB/c mice, respectively. All NA mutants (excepted D198E/N) showed RI/HRI phenotypes against ≥ 2 NAIs. These mutants grew to comparable titers of the recombinant wild-type (WT) IBV in vitro, and some of them (H136N, I222T and N294S mutants) induced more weight loss and mortality in BALB/c mice in comparison to the recombinant WT IBV. These results demonstrate that, in contemporary IBVs, some NA mutations may confer RI/HRI phenotypes to existing NAIs without altering the viral fitness. This reinforces the need for development of novel antiviral strategies with different mechanisms of action.

## 1. Introduction

Influenza B viruses (IBVs) belong to the Orthomyxoviridae family. Contrasting with influenza A viruses (IAVs), IBVs cannot be subtyped based on their hemagglutinin (HA) and neuraminidase (NA) surface glycoproteins; nevertheless, antigenic and genetic characterizations allowed their separation into two lineages represented by B/Victoria/2/1987 and B/Yamagata/16/1988 strains [1]. Humans constitute the natural and predominant host for IBVs, although a few animal species, such as seals and domestic pigs, can also be infected [2].

Each year, IAV and IBV infections result in approximately one billion infections, causing an average of 3–5 million cases of severe diseases and up to 650,000 deaths worldwide [3]. For a long time, the contribution of IBVs to such seasonal influenza burden has been underestimated because these infections generally represented a small fraction of total cases and they were considered to cause milder infections in adults compared to IAVs [4]. Contrasting with this belief, a recent study demonstrated that influenza A and B infections induced similar lengths of hospital stay, intensive care unit admissions, and rates of death in hospitalized adults [5]. Accordingly, ferret studies demonstrated that IBVs replicate efficiently in the lower respiratory tract and cause infection with the same severity as IAVs [6]. Furthermore, the rate of seasonal IBV infections may surpass that of IAV cases in some influenza epidemics. For instance, during the 2017/2018 season, 63% of all seasonal influenza cases reported in Europe were caused by IBVs [7].

The neuraminidase inhibitors (NAIs) constitute the main class of antivirals currently recommended against IAV and IBV infections. These compounds target the active site of the NA composed by 8 amino acids forming the catalytic site (R-118, D-151, R-152, R-224, E-276, R-292, R-371 and Y-406) and 11 framework residues (E-119, R-156, W-178, S-179, D-198, I-222, E-227, H274, E-277, N-294 and E-425) (N2 numbering herein and throughout the text). These residues are highly conserved among influenza A and B strains [8]. Currently, three NAIs (i.e., oseltamivir, zanamivir and peramivir) are widely available for the control of influenza infections [9]. Numerous in vitro studies and clinical reports involving IAVs and IBVs reported the emergence of strains with reduced susceptibility to NAIs, with and without being under drug pressure [10,11,12]. In most cases, IAV and/or IBV NAI-resistant variants contained amino acid substitutions within the active site of the NA enzyme [13]. Nevertheless, some mutations were also identified outside the active site involving mechanisms of resistance that remain to be clarified [14]. Noteworthily, while certain IBV NA mutations mediate a phenotype of reduced inhibition (RI; 5- to 50-fold increase in IC_50_ over the WT) or highly-reduced inhibition (HRI; ≥50-fold increase) to a specific NAI without altering the susceptibility to other inhibitors, mutations associated with a cross-resistance phenotype to at least two NAIs have also been reported in various strains belonging to different genetic lineages [15]. The eventual emergence and dissemination of such variants represent a serious clinical concern.

In past studies, some variations were observed in the susceptibility phenotypes against NAIs when assessed in different IBV strains [10,12]. Moreover, little is understood about the impact of these NA changes on in vitro and in vivo viral fitness for current IBV strains. In this report, we seek to update our knowledge of NAI susceptibility phenotypes and in vitro/in vivo viral fitness of currently circulating IBV strains harboring NA mutations, conferring potential cross-resistance phenotypes to NAIs. For that purpose, we created a reverse genetics system using the contemporary influenza B/Phuket/3073/2013 genetic background, which allowed us to rescue and characterize IBV variants containing H136N, R152K, D198E/N, I222T and N294S NA substitutions reported by WHO as clinical molecular markers of RI/HRI to multiple NAIs [16].

## 2. Materials and Methods

### 2.1. Virus and Cells

The influenza B/Phuket/3073/2013 virus (Yamagata lineage-related vaccine strain since the 2015-2016 season) was obtained from NIBSC (code number #17/248). Madin-Darby canine kidney (MDCK) and ST6GalI-MDCK cells (kindly provided by Y. Kawaoka from the University of Wisconsin, Madison, WI, USA) [17] were cultured in Minimum Essential Medium (MEM, Invitrogen, Carlsbad, CA, USA) supplemented with 10% fetal bovine serum (FBS, Invitrogen), HEPES and antibiotics. The human embryonic kidney 293T cell line (ATCC, CRL-3216) was maintained in Dulbecco’s modified Eagle’s medium (DMEM, Invitrogen), supplemented with 10% FBS and HEPES.

### 2.2. Rescue of Recombinant IBVs

To produce recombinant influenza B/Phuket/3073/2013 viruses, we used the bidirectional pBZ plasmid (kindly provided by B. Zhou from the University of New York, USA) [18]. Each viral segment was cloned into the pBZ plasmid. The pBZ-NA plasmid was used for inserting the selected NA mutations by PCR-mediated mutagenesis (Stratagene, La Jolla, CA, USA). Recombinant viruses were rescued as previously described [19], with minor modifications. Briefly, 293T/MDCK cells (5 × 10^5^ cells of each line) were co-cultured in 6-well plates with DMEM + 10% FBS + HEPES for 48 h at 37 °C. Then, co-cultured cells were transfected with 1 μg of each pBZ plasmid and lipofectamine (3 μL/μg DNA) in 1.2 mL of OptiMEM for 6 h. The medium was replaced by DMEM + 0.1% bovine serum albumin (BSA) + bovine trypsin (TPCK-treated, Sigma, St-Louis, MO, USA) and co-cultures were incubated at 33 °C for 72 h before harvesting viruses.

All IBV genomic segments within recombinant pBZ plasmids, as well as HA and NA genes of recombinant IBVs recovered during replicative capacity and mouse experiments, were sequenced using the ABI 3730 DNA Analyzer (Applied Biosystems, Carlsbad, CA, USA).

### 2.3. Phenotype of Susceptibility to NAIs and Relative NA Activity

NA inhibition assays using the methylumbelliferone N-acetylneuraminic acid (MUNANA, Sigma, St-Louis, MO, USA) substrate were performed for determining the median inhibition concentration (IC_50_) for oseltamivir carboxylate (Hoffmann-La Roche, Basel, Switzerland), zanamivir (Sigma) and peramivir (BioCryst, Birmingham, AL, USA), as previously described [20]. The phenotypes of susceptibility to NAIs for recombinant IBVs were reported according to WHO criteria: NI, normal inhibition (< 5-fold increase in IC_50_ over WT); RI, reduced inhibition (5- to 50-fold increase in IC_50_ over WT) and HRI, highly reduced inhibition (> 50-fold increase in IC_50_ over WT) [16].

To determine the impact of NA substitutions on the NA activity, recombinant NA proteins were expressed in HEK-293T cells, as previously described [20]. Briefly, confluent cells (in 12-well plates) were transfected with 500 ng of the WT or mutant NA-pBZ plasmid + lipofectamine (3 μL/μg DNA) and incubated for 6 h at 37 °C. The medium was completed with 500 μL of DMEM + 20% FBS + HEPES, and transfected cells were incubated overnight at 37 °C. Then cells were harvested with 200 μL of cold PBS + 0.02% EDTA, centrifuged (1500× *g*, 5 min at 4 °C), washed with cold PBS and suspended in 200 μL of 3.5 mM CaCl_2_ cold solution. NA activity assays were then performed as described above [20].

### 2.4. In Vitro Viral Replication Kinetics

The replicative capacities of recombinant IBVs were performed by infecting STG6GalI-MDCK cells in 12-well plates with 50 plaque forming units (PFUs)/well (M.O.I. of 0.0001 PFU/cell). Viruses were harvested at 24, 48, 72 and 96 h post-infection (p.i.) and kept frozen at −80 °C until titration by TCID_50_ using STG6GalI-MDCK cells.

### 2.5. Experimental Infection of Mice

Groups of 6- to 8-week old female BALB/c mice (Charles River, Senneville, QC, Canada) (*n* = 10), were inoculated intranasally, under isoflurane anesthesia, with 5 × 10^6^ PFUs of each recombinant IBV (this inoculum was previously determined to cause ≈ 100% of mortality in mice infected with the recombinant WT virus). A control group included 3 mice that were intranasally inoculated with saline. Mortality and weight loss were monitored daily for 10 days and the humane endpoint was determined at 20% of weight loss. On day 3 p.i., four mice per infected group were sacrificed and their lungs were collected and homogenized in 1 mL of PBS containing penicillin, streptomycin, and amphotericin B, using the Omni Tip homogenizer (OMNI International, Kennesaw, GA, USA). Cells were pelleted by centrifugation (1500× *g*, 5 min, 4 °C), and supernatants were used for determination of viral titers, expressed as TCID_50_ per lung, using ST6GalI-MDCK cells.

All animal procedures were approved by the Institutional Animal Care Committee of Laval University, according to guidelines of the Canadian Council of Animal Care (permission number: 2015063-4).

### 2.6. Statistics

Replication kinetics, mouse weight loss, and lung viral titers were analyzed with one-way ANOVA Dunnett’s multiple comparisons test between the WT and the mutant groups using GraphPad Prism 6.0 (GraphPad Software Inc., La Jolla, CA, USA).

## 3. Results

### 3.1. NAI Susceptibility Phenotypes of Recombinant B Viruses

The recombinant B/Phuket/3073/2013 WT virus and its H136N, R152K, D198E/N, I222T and N294S variants were successfully rescued. In NA inhibition assays, all NA mutants except D198E/N variants demonstrated decreased viral susceptibilities to at least two NAIs among zanamivir, peramivir, and oseltamivir (Table 1). The R152K exhibited an HRI phenotype against the three NAIs, whereas a RI phenotype against these compounds was shared by the I222T and N294S mutants. The H136N variant with an HRI phenotype against zanamivir and peramivir remained susceptible to oseltamivir. Finally, the D198E/N mutants were susceptible to oseltamivir and zanamivir, although both had a RI phenotype to peramivir. In the absence of cross-resistance, the latter viruses were not further characterized.

The NA activities of recombinant H136N, D198E, D198N and I222T NA proteins were similar to that of the WT (92%, 107%, 101% and 106%, respectively) (Table 1), while the N294S NA protein showed a slight (but statistically significant) increase of activity (108% vs the WT, *p* < 0.05). Finally, the R152K NA protein had a significantly reduced activity (60% vs. the WT, *p* < 0.001) (Table 1).

### 3.2. Replicative Capacity of NAI-Resistant IBV Recombinants

In replication kinetics experiments, the recombinant WT IBV and its H136N, I222T and N294S NA variants grew to comparable titers at different time-points (Figure 1). Viral titers ranged between 10^3.2^ and 10^4.2^ TCID_50_ at 24 h to 10^7.5^ and 10^8.2^ TCID_50_ at 72 h. By contrast, the R152K mutant showed a significant reduction of the viral titer, compared to the WT (*p* < 0.01), that was only observed at the 72-h time point p.i. (Figure 1).

### 3.3. Virulence of Recombinant IBVs in Mice

In experimental infections of BALB/c mice, the recombinant H136N, I222T and N294S viruses induced body weight losses that were greater than that of the WT group. At day 4 p.i., we recorded a 14.96% body weight loss in the WT infected group while weight loss reached 22.38% (*p* < 0.001), 19.9% (*p* < 0.05), and 21.37% (*p* < 0.01) in the H136N-, I222T-, and N294S-infected groups, respectively. Finally, only 9.1% of all body weight loss was observed in mice infected with the recombinant R152K virus (*p* < 0.01) (Figure 2A).

All mice infected with the H136N, I222T and N294S recombinants died by day 4–5 p.i., whereas mortality rates of 80% and 33% were observed in the WT and R152K groups, respectively (Figure 2B). Due to a technical problem during infection, one mouse infected with the WT virus was removed from our experiment analysis (*n* = 5 instead of 6).

As shown in Figure 2C, high viral titers (between 10^4.8^ and 10^5.7^ TCID_50_ per lung) were observed in the lungs of mice collected at day 3 p.i. Although a slightly reduced viral load was observed in the R152K infected group (10^4.8^ TCID_50_ per lung), no significant difference was observed between the WT (10^5.4^ TCID_50_ per lung) and each of the mutant groups (10^5.3^, 10^5.6^ and 10^5.7^ TCID_50_ per lung for H136N, I222T, and N294S groups, respectively). Sequencing of viral HA and NA from lung homogenates confirmed the presence of the original NA mutation without other NA or HA changes.

## 4. Discussion

The eventual emergence and dissemination of multi-NAI resistant IBVs represent a serious clinical concern. Whether baloxavir marboxil, the new anti-influenza agent targeting the PA subunit of the viral polymerase [21], could constitute a suitable alternative against NAI-resistant IBV infections remains to be investigated. Indeed, rates of baloxavir resistance close to 20% were reported in H3N2-infected patients who received baloxavir therapy [22].

In this study, we used a unique contemporary viral background to investigate the impact of NA mutations previously associated with NAIs cross-resistance phenotypes in various clinical IBV strains. The use of the influenza B/Phuket/3073/2013 strain as a backbone in our experiments is important because this strain has been the vaccine component representing the Yamagata lineage in the northern hemisphere since the 2015-2016 season [7,23,24]. As part of this study, five NA substitutions (R152K, D198E, D198N, I222T and N294S) were selected on the basis of a recent WHO update on NAI resistance in clinical IAV and IBV isolates [16]. All these mutations were reported to confer a cross-resistance phenotype to at least two NAIs in IBV isolates from the Yamagata lineage. Influenza B/Memphis/20/1996-R152K and B/Rochester/02/2001-D198N variants were described in patients who received zanamivir and oseltamivir, respectively [25,26]. Additional R152K and D198N clinical isolates were identified in surveillance programs from untreated subjects [12,15]. The N294S substitution was observed in an immunocompromised child prior to oseltamivir therapy [27], and no additional cases have been reported since then. The D198E mutation detected in the B/Perth/211/2001 variant in the absence of NAI therapy also seemed to be rare in the Yamagata lineage, contrasting with the I222T substitution that was described in a recent surveillance update [11,12]. We also included the H136N substitution in our study since this variant was detected in a B/Laos/0406/2016 virus (Victoria lineage) with a cross-resistance phenotype [28].

Our NA inhibition assays data are in agreement with those of WHO regarding NAI susceptibilities and confirmed the RI/HRI phenotype to ≥ 2 NAIs for all mutants, except for the D198E and D198N variants, that remained susceptible to oseltamivir (4- and 3-fold increase in IC_50_ values over WT, respectively) and zanamivir (4- and 3-fold increase, respectively) but had a RI phenotype against peramivir (21- and 9- fold increase, respectively). In previous studies, the D198E substitution conferred RI phenotypes to oseltamivir and zanamivir (in B/Perth/211/2001) or an RI phenotype against oseltamivir alone (in B/Yamanashi/166/1998 recombinant viruses) [11,29]. However, D198E was observed for the last time in 2001 in the Yamagata lineage, and it is possible that this substitution does not confer a RI phenotype any longer in the contemporary B/Phuket/3073/2013 viral background. The D198N susceptibility results are in agreement with recent surveillance data, where a B/Wyoming/07/2017 (B/Yamagata lineage) variant had an RI phenotype to peramivir, was susceptible to zanamivir and had a borderline RI phenotype to oseltamivir (5.8-fold increase of IC_50_ value compared to WT) [15]. In addition, our N294S recombinant exhibited an RI phenotype to zanamivir, whereas this mutant was previously described as susceptible [29]. Thus, the susceptibility phenotype for some NA substitutions may vary depending on the viral genetic backgrounds. In addition, the viral evolution may introduce NA substitutions whose accumulation over time may interfere with the susceptibility results when associated with NAI-resistance mutations. For example, the role of the D198N NA substitution on RI/HRI phenotypes could not be confirmed in the B/Singapore/GP702/2015 virus [30]. Thus, based on these observations, genotypic characterization of clinical isolates may not be sufficient to determine the susceptibility of a specific variant, and phenotypic testing could still be required.

The impact of NA mutations on the in vitro viral fitness of our recombinants viruses was further assessed. We did not observe a significant difference between viral titers of the mutants compared to the WT, except at the 72 h p.i. time-point where reduced titers were observed for the R152K mutants (*p* < 0.01 vs WT) (Figure 1). Of note, viral replication of our recombinant H136N mutant was not impaired while this substitution altered the fitness of B/Laos/0406/2016 virus [28]. The difference in the lineage between the two strains (Yamagata for our strain vs Victoria for B/Laos/0406/2016) could be responsible for such discrepancy.

Besides in vitro properties, we also seek to characterize our recombinant viruses using an in vivo system. Although the mouse model does not constitute the gold standard for studying the replication and transmission of influenza B viruses to the same extent as ferrets, it may provide a technical and cost-favorable option for evaluating some experimental parameters, such as weight loss, mortality rates, and lung viral titers. However, mice are not naturally susceptible to IBV infections, and multiple lung-to-lung passages resulting in numerous HA/M/NP/PA mutations are usually needed to adapt a specific IBV virus strain to that animal species [31]. Interestingly, the B/Phuket/3073/2013 vaccine strain and its recombinant virus were found to be lethal in BALB/c mice without prior adaptation, when using a high viral inoculum (i.e., 5 × 10^6^ PFUs). Indeed, no mortality could be observed when we used a lower inoculum of 5 × 10^5^ PFUs which is in agreement with a previous study [32]. Noteworthily, the characterization of the viral fitness in our model does not take into account the potential effect of mutations that could emerge during the mouse adaptation process [31]. As the original B/Phuket/3073/2013 vaccine strain was passaged in embryonated eggs, we identified in its HA protein the N196D egg-adapted substitution that could potentially increase viral pathogenicity in mice through adaptation to a α2,3 sialic acid receptor. A similar role was previously attributed for the S190R A(H1N1) egg-adapted HA mutation [33,34].

In our study, all mice infected with H136N, I222T and N294S recombinant viruses died, contrasting with the R152K recombinant virus that killed only 33% of the mice (Figure 2B). Thus, weight loss and mortality data demonstrated that in vivo viral fitness was not altered in multi-resistant NA mutants (except for R152K). Although the in vitro replicative capacity of R152K variant was not impaired, the in vivo-reduced viral fitness for this variant could be explained by the fact that the R152 is a catalytic residue contrasting to the remaining mutations, which are rather framework substitutions (Table 1). In addition, we showed that, in contrast to the other mutant NAs, the recombinant R152K NA enzyme had lower activity when compared to that of the WT NA (Table 1). A previous study demonstrated that the influenza NA activity has a more important role in vivo than in MDCK cell-culture context due to the presence of a mucin/surfactant-rich environment in the respiratory tract [35]. The ability of an influenza virus to disseminate through the mucus barrier is dependent on its NA activity that prevents viruses from being trapped [36].

## 5. Conclusions

In conclusion, our approach using a contemporary and pathogenic IBV recombinant strain confirms that most NA substitutions associated with NAI resistance in previous strains still confer a cross-resistance phenotype in the B/Phuket/3073/2013 background. However, we noted some discrepancies with regard to the D198E/N variants, highlighting the potential variability of NAI susceptibility results between different strains. More interestingly, we demonstrated that some NAI-resistant IBV strains could retain viral fitness in vitro and in a mouse model. The impact of these mutations on viral transmission remains to be clarified by further experiments in ferrets. These findings warrant continuous surveillance programs and reinforce the need to develop novel antiviral strategies.

## Figures and Tables

**Figure 1 viruses-11-00006-f001:**
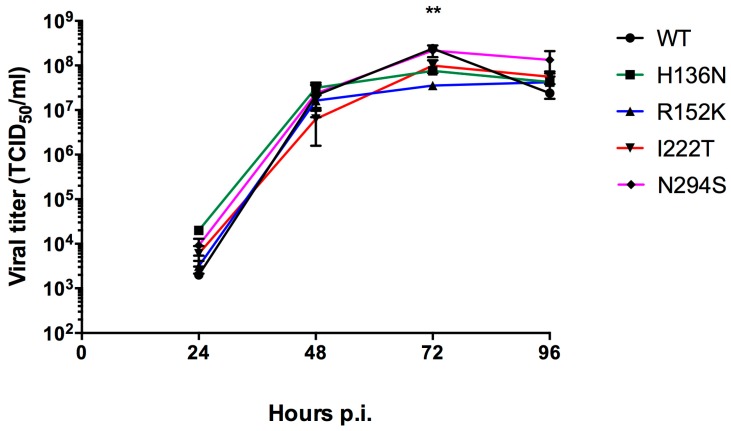
Replicative capacity of recombinant influenza B/Phuket/3073/2013 viruses in STG6GalI-MDCK cells. Cells were infected with a MOI = 0.0001 in triplicate for each group and viruses were harvested at 24, 48, 72 and 96 h p.i. and frozen at −80 °C before viral titration by TCID_50_. **, *p* < 0.01 between WT and R152K groups using one-way ANOVA Dunnett’s multiple comparisons of each mutant relative to the WT virus.

**Figure 2 viruses-11-00006-f002:**
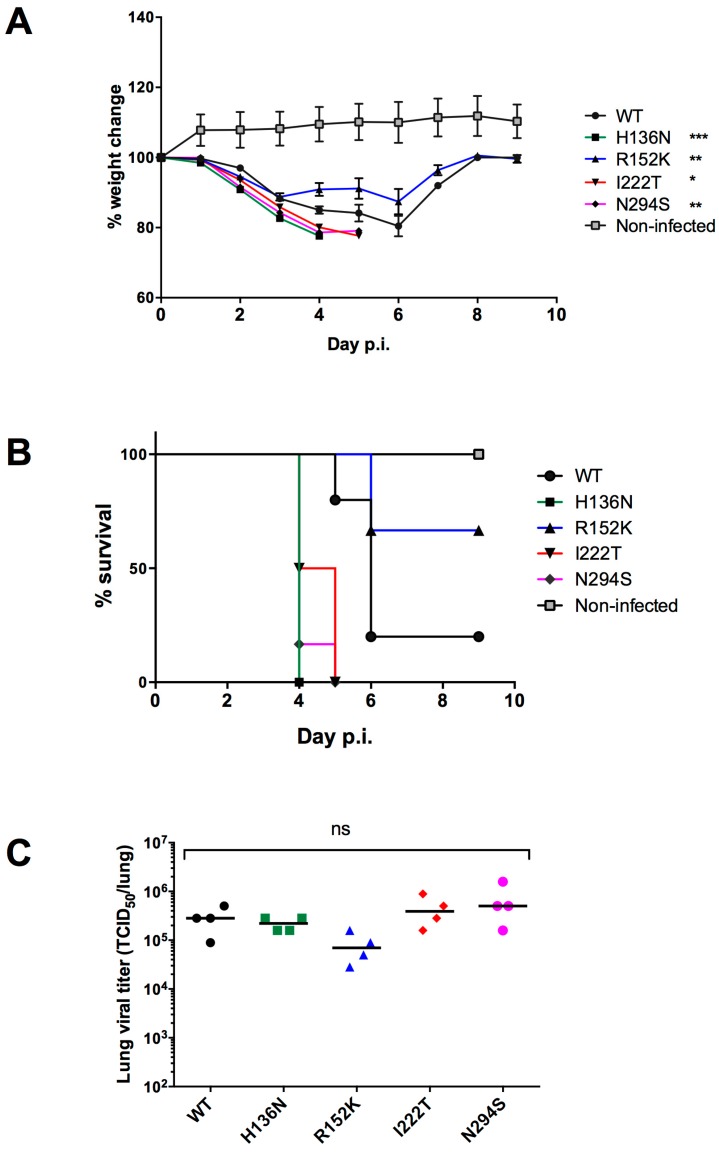
Virulence of recombinant IBVs with reduced susceptibility to NAIs. In each group, ten BALB/c mice were infected by the intranasal route with 5 × 10^6^ PFUs of each recombinant B/Phuket/3073/2013 virus. Weight loss (**A**) and mortality (**B**) were monitored throughout the experiment. At day 3 p.i., 4 mice in each group were sacrificed and their lungs were harvested to determine lung viral titers (**C**). Statistical analyses of weight loss at day 4 (indicated in the legend) and lung titers were performed using one-way ANOVA, Dunnett’s multiple comparisons test between the WT group and each of the mutant groups. ns, not significant, *, *p* < 0.05 between WT and I222T, **, *p* < 0.01 between WT and R152K or N294S groups, ***, *p* < 0.001 between WT and H136N groups. For the WT group, due to a technical problem during infection, one mouse was excluded from our analysis.

**Table 1 viruses-11-00006-t001:** Phenotype of resistance to neuraminidase (NA) inhibitors for recombinant influenza B/Phuket/3073/2013 viruses.

	Neuraminidase Inhibitors IC_50_ values (nM) ^a^															NA Activity ^b^ (%)		
Substitution^c^	Oseltamivir			Folds^d^	Phenotype*^e^*	Zanamivir			Folds^d^	Phenotype^e^	Peramivir			Folds*^d^*	Phenotype*^e^*			
WT	16.98	±	5.11	1		0.76	±	0.12	1		0.74	±	0.13	1		100	±	1.87
H136N	57.20	±	11.41	3	NI	161.22	±	23.47	212	HRI	177.46	±	199.9	240	HRI	92	±	6.69
R152K	1017.79	±	108.71	60	HRI	63.83	±	2.79	84	HRI	422.63	±	45.98	572	HRI	60^***^	±	2.0
D198E	75.89	±	5.89	4	NI	2.98	±	1.05	4	NI	15.53	±	0.90	21	RI	107	±	3.4
D198N	52.30	±	2.87	3	NI	2.23	±	0.17	3	NI	6.92	±	0.36	9	RI	101	±	0.44
I222T	111.25	±	37.33	6	RI	3.90	±	1.61	5	RI	19.48	±	5.91	26	RI	106	±	3.32
N294S	178.85	±	94.09	10	RI	13.97	±	8.25	18	RI	33.68	±	18.98	46	RI	108^*^	±	0.95

^a^ Means IC_50_ values +/- SD were obtained from two independent experiments in duplicate; ^b^ NA activity was obtained from triplicate of a single experiment and results were standardized compared to the WT (%); *, *p* < 0.05 compared to the WT; ***, *p* < 0.001 compared to the WT; ^c^ Substitutions are reported in N2 numbering; ^d^ Fold changes according to WT virus; ^e^ Resistance phenotype in accordance with WHO criteria: NI, Normal Inhibition (≤5-fold increase for influenza B); RI, Reduced Inhibition (between 5- and 50-fold increase for influenza B); HRI, Highly Reduced Inhibition (>50-fold increase for influenza B).

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
