# Peer review of "In Vitro Properties and Virulence of Contemporary Recombinant Influenza B Viruses Harboring Mutations of Cross-Resistance to Neuraminidase Inhibitors"

_viruses, 2018, doi:10.3390/v11010006_

Reviewer 1 Report

In this manuscript by Fage et al., the authors investigated the impact of some amino-acid substitutions that, in influenza B viruses, are associated with resistance to neuraminidase inhibitors.Using reverse genetics, they rescued one “wild-type” influenza B virus isolate corresponding to the contemporary vaccine strain of the Yamagata lineage (B/Phuket/3073/2013), along with six variants harbouring substitutions in the neuraminidase that were reported as molecular markers of reduced inhibition (RI) or highly reduced inhibition (HRI) to multiple neuraminidase inhibitors (NAIs). The seven recombinant viruses were then evaluated for (i) their NAI susceptibility, (ii) their replication kinetics in cultured cells and (iii) their phenotype in experimentally infected mice.In an in vitro assay with synthetic substrate MUNANA, four of the mutant viruses investigated (H136N, R152K, I222T and N294S) showed decreased susceptibility to at least two of the three NAIs (zanamivir, oseltamivir and peramivir). The R152K mutant showed a HRI phenotype against all three NAIs.These four mutants showed unaltered replication kinetics in STG6GalI-MDCK cells, except for a slightly reduced titer observed at 72h p.i. for the R152K mutant.Three of the above-mentioned mutants exhibited an increased pathogenicity in mice (H136N, I222T and N294S) relative to the WT virus, while the R152K mutant showed a reduced pathogenicity, as judged from the body weight loss and the survival curves. However, viral loads at 3 days p.i. in the lungs of infected mice showed no significant differences between the five viruses(wt and four mutants) that were investigated in vivo.This is an interesting manuscript, however there are a number of points that should be addressed by the authors to improve its quality.

Major remarks

Lines 126-134 and Table 1. The authors characterized the NAI susceptibility of the wt and mutant viruses, but did not show the control NA activity of their viruses (i.e. in vitro activity in the absence of NAIs). This information is required, since perhaps it could help understand the pathogenicity phenotype of the mutants.

Figure 1. The viral titers should be displayed as line-connected dots rather than bars, in order to emphasize the continuity of the time-based kinetics. Further, for the sake of readability, values (not their log10) should be reported on a graph with a log10 y-scale, with subdivisions [2-9] and powers of 10 (10^2, 10^4 etc).

Line 152. ..Dunnet’s multiple comparisons of each mutant relative to the WT virus.

Line 147. …reduction of the viral titer, compared to the WT (p<0.01), that was observed only at the 72h time point.

Lines 155-159. Re-order the sentences or the informations. For instance “At 4 days p.i. we recorded a 15% body weight loss in the WT group. By contrast, the body weight loss reached 22%, 20% and 21% in the H136N, I122T and N249S groups, respectively, while it reached only 9% in the R152K group.”Line 161. … susceptibility to NAIs. In each group, ten BALB/c mice were…

Line 166. … ANOVA, Dunnett’s multiple comparisons between the WT group and each of the mutant groups.

Figure 2C (and line 118). First, viral loads should be expressed as TCID50 per lung, or per mg of lung. Then, these viral loads (not their log10 value) should be reported on a graph with a log10 y-scale with subdivisions [2-9] and with powers of 10 (10^2, 10^4 etc). For the sake of readability, groups of values should be displayed as individual values with only the median (not the S.D.).

Lines 175-77. Reformulate the sentence. For instance “Although a slightly reduced viral load was recorded in the R152K group, no significant differences were observed between groups.” As for the statistical significance, it depends on the number of comparisons that were performed in the Dunnett’s test. The authors are imprecise about the number of comparisons that were made: are there the ten possible comparisons, or only the four WT vs mutant comparisons? Depending on that, the sentences above could be modified accordingly (“no significant differences .. between groups” changed to “no significant differences … between the WT and each of the mutant groups”).

The discussion is poorly informative, and should be shortened.

Lines 216-19 and 227-231, the authors should be more precise about the other possible substitutions (in NA, but also possibly in HA or in M1) that could modulate the impact of the identified substitutions on the resistance or susceptibility to NAIs (rather than the vague sentences in lines 215-219). The authors should also discuss and propose hypotheses for the increased pathogenicity of the three mutants (H136N, I222T and N294S), and the reduced pathogenicity of the R152K.

Lines 242-44. The sentence and its meaning are unclear.

Lines 245-47. From the sentence it is unclear whether the vaccine strain B/Phuket/3073/2013 was demonstrated to harbour the N196D substitution in its HA.

Minor remarks

Lines 181-82. …targeting the viral polymerase.. (or the PA subunit of the viral polymerase). Lines 185-186. .the impact of NA mutations observed in clinical samples and associated with cross-resistance phenotypes (clinical mutation is meaningless). Lines 213-14. …had a borderline RI phenotype (5.8-fold increase of IC50) to oseltamivir. Lines 220-21. ..observations, genotypic characterization of clinical isolates is not sufficient … and phenotypic testing is still required. Lines 240-41. Indeed, we observed no mortality when we inoculated mice with 5x10^5 PFUs (dat not shown), in agreement with a previous study where…

Author Response

Major remarks

Lines 126-134 and Table 1. The authors characterized the NAI susceptibility of the wt and mutant viruses, but did not show the control NA activity of their viruses (i.e. in vitro activity in the absence of NAIs). This information is required, since perhaps it could help understand the pathogenicity phenotype of the mutants.

- As suggested, an additional experiment was performed for the determination of NA activity using recombinant NAs (please see, materials and methods: lines 103-110). The related results are summarized in the last column of Table 1 and in the results section (lines 146-149).

Figure 1.  The viral titers should be displayed as line-connected dots rather than bars, in order to emphasize the continuity of the time-based kinetics. Further, for the sake of readability, values (not their log10) should be   reported on a graph with a log10 y-scale, with subdivisions [2-9] and   powers of 10 (10^2, 10^4 etc).

- Figure 1 was modified as suggested and colors were added to improve the readability.

Line 152. Dunnet’s multiple comparisons of each mutant relative to the WT virus.

- This was corrected: Lines 172-173.

Line 147. …reduction of the viral titer, compared to the WT (p<0.01), that  was observed only at the 72h time point.

- The statement was changed as: By contrast, the R152K mutant showed a significant reduction of the viral titer, compared to the WT (P < 0.01), that was only observed at the 72h p.i. time point (figure 1) (lines 165-167).

Lines 155-159. Re-order the sentences or the informations. For instance “At  4 days p.i. we recorded a 15% body weight loss in the WT group. By contrast, the body weight loss reached 22%, 20% and 21% in the H136N, I122T and N249S  groups, respectively, while it reached only 9% in the R152K group.”

- The sentences were re-ordered as: At day 4 p.i., we recorded a 14.96% body weight loss in the WT infected group while weight losses reached 22.38% (P < 0.001), 19.9% (P < 0.05) and 21.37% (P < 0.01) in the H136N-, I222T- and N294S-infected groups respectively. Finally, only 9.1% of body weight loss was observed in mice infected with the recombinant R152K virus (P < 0.01) (figure 2A) (lines 176-180).

Line 161. … susceptibility to NAIs. In each group, ten BALB/c mice were…

- This was corrected: Lines 182-183.

Line 166. … ANOVA, Dunnett’s multiple comparisons between the WT group and each of the mutant groups.

- This was corrected: Lines 187-188.

Figure 2C (and line 118). First, viral loads should be expressed as TCID50 per lung, or per mg of lung. Then, these viral loads (not their log10 value) should be reported on a graph with a log10 y-scale with subdivisions [2-9] and with powers of 10 (10^2, 10^4 etc). For the sake of readability, groups of values should be displayed as individual values with only the median (not the S.D.).

-  Viral titers are now expressed as TCID50 per lung in the text (lines 127; 197-200) as well as in Figure 2C. Also, colors were added in figure 2A/B/C for clarity.

Lines 175-77. Reformulate the sentence. For instance “Although a slightly reduced viral load was recorded in the R152K group, no significant differences were observed between groups.

- This statement was reformulated as: Although a slightly reduced viral load was observed in the R152K infected group (104.8 TCID50 per lung), no significant difference was observed between the WT (105.4 TCID50 per lung) and each of the mutant groups (105.3, 105.6 and 105.7 TCID50 per lung for H136N, I222T and N294S groups respectively) (Lines 197-200).

As for the statistical significance, it depends on the number of comparisons that were performed in the Dunnett’s test. The authors are imprecise about the number of comparisons that were made: are there the ten possible comparisons, or only the four WT vs mutant comparisons? Depending on that, the sentences above could be modified accordingly (“no significant differences … between groups” changed to “no significant differences … between the WT and each of the mutant groups”).

- This was corrected as indicated above.

The discussion is poorly informative, and should be shortened.

- We deleted some sentences and reformulated others for more clarity.

Lines 216-19 and 227-231, the authors should be more precise about the other possible substitutions (in NA, but also possibly in HA or in M1) that could modulate the impact of the identified substitutions on the resistance or susceptibility to NAIs (rather than the vague sentences in lines 215-219).

- The resistance phenotype determined in vitro by NA inhibition assays is exclusively dependent on the NA enzyme. Changes in the HA or M1 genes are unlikely to influence NAI susceptibility tests. Nevertheless, changes in the HA, M1 and other genes could affect viral fitness of susceptible or resistant viruses. For instance, changes at these genes could be required for mouse adaption of IBV as indicated now (lines 257-259).

The authors should also discuss and propose hypotheses for the increased pathogenicity of the three mutants (H136N, I222T and N294S), and the reduced  pathogenicity of the R152K.

- Some statements were added in the discussion: Line 272-280.

Lines 242-44. The sentence and its meaning are unclear.

- Lines 257-259 were added and lines 263-265 were reformulated as follows: Indeed, no mortality could be observed when we used an inoculum of 5x105 PFUs (data not shown) which is in agreement with a previous study [32]. Noteworthy, the characterization of the viral fitness with our model doses not take into account the potential effect of mutations that could emerge during the mouse adaptation process [31].

Lines 245-47. From the sentence it is unclear whether the vaccine strain B/Phuket/3073/2013 was demonstrated to harbour the N196D substitution in its HA.

- Indeed, we sequenced the HA gene of the B/Phuket/3073/2013 vaccine strain and identified this mutation. This was clarified in the text: Line 266.

Minor remarks

Lines 181-82. …targeting the viral polymerase.. (or the PA subunit of the  viral polymerase).

- This was corrected: Lines 204-205.

Lines 185-186. ..the impact of NA mutations observed in clinical samples  and associated with cross-resistance phenotypes (clinical mutation is  meaningless).

- This was changed as: the impact of NA mutations previously associated with NAIs cross-resistance phenotypes in various clinical IBV strains (lines 209-210).

Lines 213-14. …had a borderline RI phenotype (5.8-fold increase of IC50)  to oseltamivir.

- This was corrected: Lines 236-237.

Lines 220-21. ..observations, genotypic characterization of clinical  isolates is not sufficient … and phenotypic testing is still required.

- This was corrected: Lines 243-245.

Lines 240-41. Indeed, we observed no mortality when we inoculated mice with 5x10^5 PFUs (dat not shown), in agreement with a previous study where…

- This was reformulated as: Indeed, no mortality could be observed when we used an inoculum of 5x105 PFUs (data not shown) which is in agreement with a previous study [32] (Lines 261-265). 

Reviewer 2 Report

The manuscript is a very clear and sound presentation of data about resistance of different IBV strains against antiviral drugs with special regards to cross resistance; its a very interesting work, definitely relevant to scientists in the field; presentation, figures and tables are adequate.

minor comment: the term "recombinant" might be a bit confusing, apparently it is used to describe the viruses generated by reverse genetics, however e.g. in line 144 "the recombinant WT-IBV" is confusing, although its made by recombinant methods, the resulting virus is the WT-virus;

Overall, the manuscript is highly suited for publication,

Author Response

Minor comment

The term «recombinant» migh be a bit confusing, apparently it is used to describe the viruses generated by reverse genetics. However e.g. in line 144 «the recombinant WT-IBV» is confusing, although its made by recombinant methods, the resulting virus is the WT-virus.

- We prefer to keep the term «recombinant» when we talk about viruses produced by reverse genetics (WT or mutants).

Round  2

Reviewer 1 Report

The authors have adequately responded to the points I raised, and have modified their manuscript accordingly. However, some minor points should be considered.

Lines 57 and 263 : Noteworthily...

Line 264. .. does not take into...

Figure 1: the y-scale could begin at 10^2 (same remark for Fig 2C), and the y-scale label should be completed (TCID50 per ml, I suppose).

Author Response

The authors have adequately responded to the points I raised, and have modified their manuscript accordingly. However, some minor points should be considered.

Lines 57 and 263 : Noteworthily...

- This was modified. Lines 57 and 265.

Line 264. .. does not take into... Line 266

- This was corrected.

Figure 1: the y-scale could begin at 10^2 (same remark for Fig 2C), and the y-scale label should be completed (TCID50 per ml, I suppose).

- Figure 1 and 2C were modified as suggested and TCID50/ml was added in figure 2C.